# Impact Assessment of Urban Flood on Traffic Disruption using Rainfall–Depth–Vehicle Speed Relationship

**Kyung-Su Choo** **, Dong-Ho Kang and Byung-Sik Kim \***

Department of Urban and Environmental and Disaster Management, Graduate School of Disaster Prevention, Kangwon National University, Kangwon-do 25913, Korea; Chu_93@kangwon.ac.kr (K.-S.C.); kdh@kangwon.ac.kr (D.-H.K.)
\* Correspondence: hydrokbs@kangwon.ac.kr; Tel.: +82-33570-6819

**Abstract:** The transportation network enables movement of people and goods and is the basis of economic activity. Recently, short-term locally heavy rains occur frequently in urban areas, causing serious obstacles to road flooding and increasing economic and social effects. Therefore, in advanced weather countries, many studies have been conducted on realistic and reliable impact forecasting by analyzing socioeconomic impacts, not just information transmission as weather forecasts. In this paper, we use the Spatial Runoff Assessment Tool (S-RAT) and Flood Inundation model (FLO-2D model) to calculate the flooding level in urban areas caused by rainfall and use the flooding rate. In addition, the rainfall–flood depth curve and the Flood–Vehicle Speed curve were presented during the analysis, and the traffic disruption map was prepared using this. The results of this study were compared with previous studies and verified by rainfall events in 2011. As a result of the verification, the result was similar to the actual flooding, and when the same rainfall occurred within the range of the target area, it was confirmed that there were sections that could not be passed and sections that could be passed smoothly. Therefore, the results suggested in this study will be helpful for the driver's route selection by using the urban flood damage analysis and vehicle driving speed analysis.

**Keywords:** urban flood; rainfall (mm)–depth (m)–vehicle speed (km/h) relationship; vehicle speed reduction; traffic disruption; impact forecasting

## 1. Introduction

Climate change is expected to increase the frequency and intensity of rainfall worldwide [1,2], causing further domestic flood damage to urban areas following rapid urbanization and industrialization [3]. This calls for the consideration of climate change impact assessment in urban planning [4]. The World Meteorological Organization (WMO) emphasizes the need for forecasting impacts that take into account the socioeconomic impacts that may arise from meteorological phenomena [5] and recognizes that the Meteorological Agency needs to establish a system for an effective response to meteorological disasters [6]. Developed countries are now able to analyze and provide high-resolution weather information as well as related socioeconomic impacts. For example, the UK Flood Forecasting Center (FFC)-Flood Guidance Statement (FGS) assesses the risk for all flood types over five days, with the results provided daily [7]. In particular, it found that the degradation of the transportation system caused by urban flooding is the most detrimental to society and is estimated to be about British Pound (GBP) 100 thousand per hour for each major road affected [8–10]. As such, rainfall is a factor that negatively affects road traffic, but in Korea, it provides weather information as well as traffic information but it relies solely on wide-area forecasts and does not convey its effects

properly [11–13]. Currently, the Road Traffic Act does not provide a standard for the speed of weather changes and studies on rainfall and transportation are still insufficient [14].

In the recent trend of flood-induced research, in Kang et al. (2007) [15], the flood analysis model, such as the FLUvial Modeling Engine (FLUMEN), is used to present the flood damage reduction measures with the model after actual cases and verification. It is suggested that the actual inundation trace and the model inundation range showed a 0.16 km$^2$ difference. Simões et al. [16] analyzed the flood modeling using Infoworks model, assessed the degree of flooding in the same event of rainfall events, and suggested that the ultimate goal is to generate a probabilistic flood map. This required several factors to be considered; in Li et al. [17], the Xin'anjiang (XAJ) models developed in China and Soil and Water Assessment (SWAT) models effective in flood simulation were compared. The results showed that the XAJ model had a better performance than the sub-daily SWAT model regarding relative runoff error (RRE), but the SWAT model performed well, according to the relative peak discharge error (RPE) and error of occurrence time of peak flow (PTE). In conclusion, the SWAT model is proposed to be used promisingly in flood simulations. Wei et al. [18] analyzed using the CityDrain3 modeling tool developed at the Unit of Environmental Engineering at the University of Innsbruck by Gregor Burger and his colleagues [19], which could include urban drainage systems. Based on this, the sponge folder model was developed and calibrated to verify the target area. This was used to calculate the flood depth, and it was possible to confirm the trend of increasing flood depth with increasing rainfall. Park and Lee [20] classified 4 grades for urban flood risk assessment and calculated them by considering various indices. The land use classification map is presented based on the fuzzy theory of the high vulnerability area of the cell at 100 m × 100 m, and the urban flood analysis is based on the Hydrologic Engineering Center's River Analysis System (HEC-RAS). Liao et al. [21] The SOBEK (named after the ancient Egyptian crocodile river god [22]) model developed in Netherlands was used to model the urban inundation and divided rivers, urban sewers, and surface runoff basins. Depths above 0.1 m were considered floods, and the actual damage information was analyzed using a social media analysis system if the sensor could not detect it. Jia et al. [23] introduced and verified China flash flood hydrological Model (CNFF-HM) developed in chinawith the theme of flash flooding. The results showed that the reservoir had a significant effect on the model simulation and that the value was dependent on the soil layer and various parameters in the reservoir. We concluded that more data is needed to represent the physical mechanisms, excluding temperature, rainfall, evaporation, and penetration that make up the model.

For research trends related to rainfall and vehicle speed, Jeong et al. [24] analyzed traffic characteristics depending on precipitation using Road Weather Information System (RWIS) and detector data and suggested that speed and traffic volume decreased during rainy periods compared to during no-rain periods. Along with an equation for speed estimation depending on rainfall intensity, they suggested that, when classifying the levels of rainfall using speed decrement according to precipitation, the criterion for distinguishing light rain and moderate rain is 0.4 mm/5 min and that for moderate rain and heavy rain is 0.8 mm/5 min. William et al. [25] proposed speed–vehicle density under various rainfall conditions considering traffic vehicle speed, speed depending on traffic volume, and traffic volume that a road can accommodate and suggested a functional formula of vehicle travel speed and traffic volume depending on rainfall intensity. They suggested that, especially in Hong Kong and other cities in Asia, rainfall intensity should be considered in the design, operation, and evaluation of road facilities because annual rainfall intensity is relatively high in these cities. Mashros et al. [26] estimated the effects of rainfall on travel speed and degree of reduction in speed using data of three months. They divided rainfall intensity into four levels and graph reduction of traffic speed at the 15th, 50th, and 85th percentiles according to rainfall intensity increase. Lu et at. [27] made predictions based on the quantitative association of rainfall intensity and travel time on the expressway using the impulse response function. He presented a graph quantitatively predicting the increase of travel time in the event of rain during evening hours when the roads are crowded with vehicles, concluded that additional correction was still needed, and pointed out the limitation of application only at certain

places. Kim and Oh [12] conducted research on changes in daily traffic volume (on weekdays and weekends) depending on rainfall intensity using data on traffic volume of ordinary national highways and rainfall data of Korea Meteorological Administration (KMA) and suggested that average daily traffic volume with rainfall increase showed differences up to 5.48% over all the days. Leong et al. [28] indicated that, although it is natural that speed decreases with rainfall, the amount of reduction increases further with an additional increase in rainfall intensity and analyzed the impact on rainfall intensity using low-resolution data. Jia et al. [29] suggested Recurrent neural network-Long short term memory (R-LSTM) and Recurrent neural network-Deep belief network (R-DBN) for prediction of short-term traffic speed considering the impact of rainfall, analyzed whether deep learning data input to outperforms the ordinary model, found that R-LSTM was the most accurate among many statistical techniques, and presented a graph showing traffic volume decrease according to rainfall intensity. They suggested that many environmental factors should be considered in a future study to attain further improvement of accuracy

Simões et al. [16] used the integrated watershed model similar to the model used in this study, only differing in its use of drainage. In the study by Li et al. [17], by comparing the SWAT model and the XAJ model, the SWAT model performed poorly in low flow rates and emphasized the necessity of supplementation. This point should be supplemented by referring to the FLO-2D model used in this study. In the case of urban planning, the flood risk grade was divided by the Hydrologic Engineering Center's River Analysis System (HEC-RAS) only [20]. Meanwhile, in Jia et al. [23], the China flash flood hydrological model (CNFF-HM) was used. The hydrological distribution model with physical mechanisms included basic data, such as temperature, rainfall, evaporation, penetration, etc. In addition, it is supposedly suitable for large-scale research because of the observation error and is difficult to apply to the stable watershed. In this study, results were derived using the coupling method of the rainfall–runoff model and the flood model, which is judged to be different from other studies.

Therefore, this study revealed the necessity of presenting the depth of urban flooding and reflecting high-resolution analyses and hydrological factors in each region to cope with the damage that may affect the vehicle and to investigate the degree of flooding and the impact on vehicle operation. The actual case was verified using Social networking service (SNS) and news data. Furthermore, this study's results were used to create a traffic disruption map using reduced vehicle speed of each cell, which can guide vehicles for their destinations by bypassing roads expected to be flooded.

## 2. Methodology and Materials

### 2.1. Methodology

In the calculation of threshold rainfall, the two models were coupled to estimate the threshold rainfall that causes the specific inundation depth of each cell (20 cm for vehicles) [30]. Subsequently, the flood depth–vehicle speed curve was prepared using previous research data [31], and the traffic disruption map was created after calculating the vehicle speed. Afterward, the traffic obstacle impact assessment was compared with the results of previous studies in Korea and verified with the damage to heavy rain on July 2011. After calculating the rainfall–depth curve, vehicle driving speed is calculated through the flood depth–vehicle speed curve and the flooding depth generated by each cell. The use of flood depths, which are commonly associated with two curves, can be used to reflect the hydrologic characteristics of each cell, which can be distinguished from previous studies. Figure 1 shows the flowchart of this study's processes, while Figure 2 shows the relationship between the curves presented in this paper and the vehicle speed in rainfall.

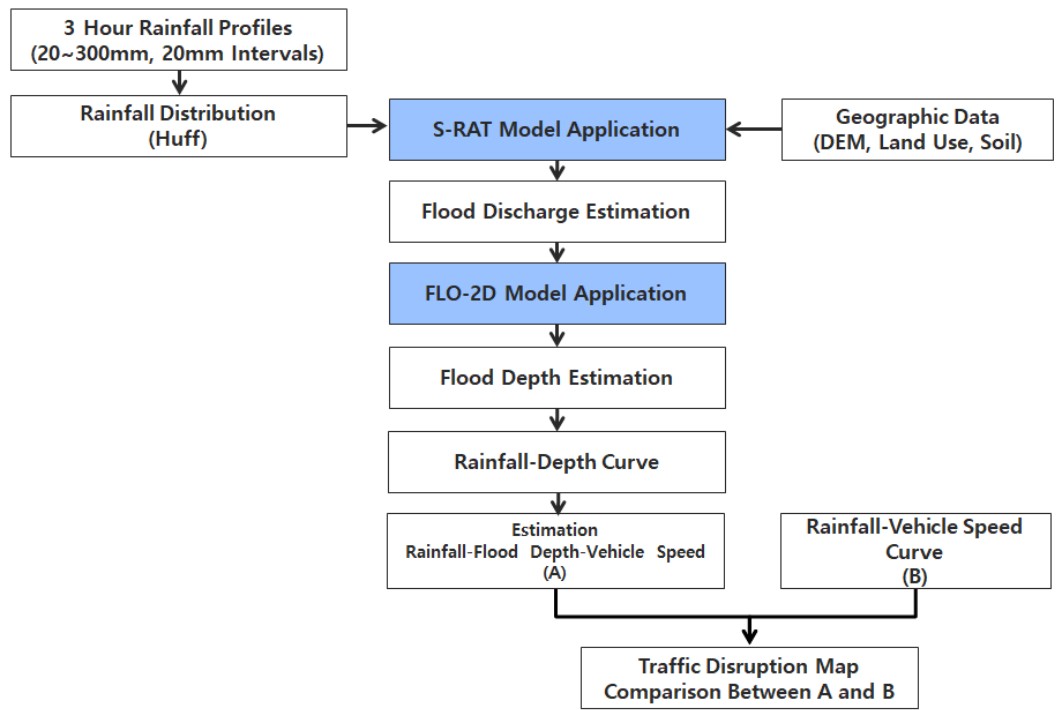

**Figure 1.** Flowchart of the study.

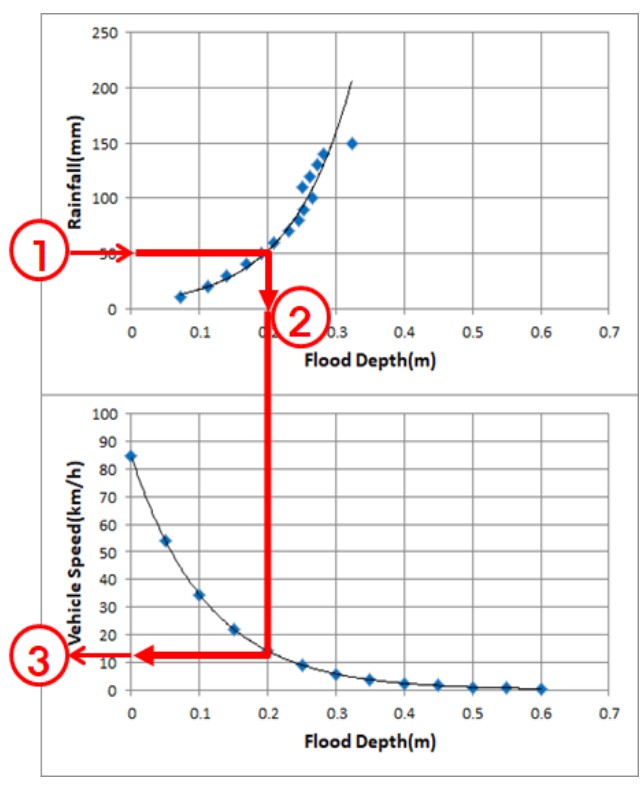

**Figure 2.** Rainfall–depth–vehicle speed relationship.

## 2.2. Distributed Rainfall–Runoff Model (S-RAT)

The S-RAT (Spatial Runoff Assessment Tool) model is a distributed rainfall–runoff model developed by Reference [32]. According to its design, a target basin is constructed with cells of fixed size using Geographic Information System GIS data, and the conceptual water balance of each cell is calculated

according to intervals of time, thereby simulating the temporal and spatial changes of discharge at the basin. S-RAT extracts parameters, and input data is simplified. The S-RAT model uses the SCS curve number (hereinafter "Runoff Curve Index (CN)" [33]) method to calculate infiltration or direct runoff for each cell. For this, a soil map and a land-use map are inputted to generate and calculate the cell data of CN values.

$$S(i,j) = 254(\frac{100}{CN(i,j)} - 1) \tag{1}$$

where $S(i,j)$ is potential retention and $CN(i,j)$ is the CN value for a cell.

$$(\frac{P_n[t,(i,j)]}{P[t,(i,j)]}) = (\frac{F[t,(i,j)]}{HS(i,j)}) \tag{2}$$

where $F[t(i,j)]$ is the water content (mm) of an infiltration basin and $HS(i,j)$ is the capacity of the infiltration basin.

$$I[t,(i,j)] = P[t,(i,j)] - P_n[t,(i,j)] \tag{3}$$

$$W[t,(i,j)] = \frac{F[t,(i,j)]}{H_s} \tag{4}$$

$$I[t,(i,j)] - W[t(i,j)] = (\frac{dF[t,(i,j)]}{dt}) \tag{5}$$

where $W[t(i,j)]$ is subsurface runoff, $P[t,(i,j)]$ (mm) is direct runoff, and $H_s$ is a conceptual dimensionless constant. The governing equation for mass conservation of the infiltration basin is calculated using Equations (3)–(5), as below. Equation (6) can be analyzed using the Runge–Kutta 4th-order method.

$$\frac{dF[t,(i,j)]}{dt} = -\frac{F[t,(i,j)]}{H_s} - E[t,(i,j)] + P[t,(i,j)]\left\{1 - \frac{F[t,(i,j)]}{HS(i,j)}\right\} \tag{6}$$

Figure 3 shows the conceptual diagram of cell water budget for the model. Discharge is calculated by inserting rainfall obtained from the above equation into S-RAT, a distributed runoff model.

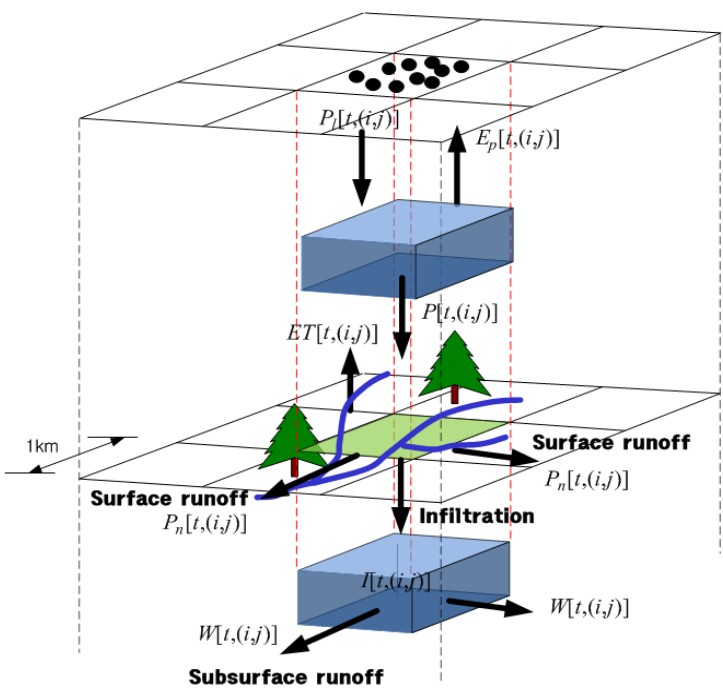

**Figure 3.** Conceptual diagram of cell water balance [32].

### 2.3. Flood Inundation Model (Flo-2D)

The FLO-2D model is a flood inundation model of volume conservation. It is a model of high reliability authorized by the US Federal Emergency Management Agency (FEMA) and can discriminate inundated areas automatically with GDS and MAPPER++.

The flow equation is shown in Equation (7) and is based on the continuity and momentum equations. Figure 4 is a schematic of the flow equation.

$$\frac{\partial h}{\partial t} + \frac{\partial hV}{\partial x} = iS_f = S_0 - \frac{\partial h}{\partial x} - \frac{V}{g}\frac{\partial V}{\partial x} - \frac{1}{g}\frac{\partial V}{\partial x} = 0 \tag{7}$$

where $h$ is the depth of discharge, $V$ is the mean velocity of one of eight directional flows ($x$ direction) at the beginning cell, $i$ is excessive rainfall intensity, and $S_f$ is friction slope based on the Manning equation.

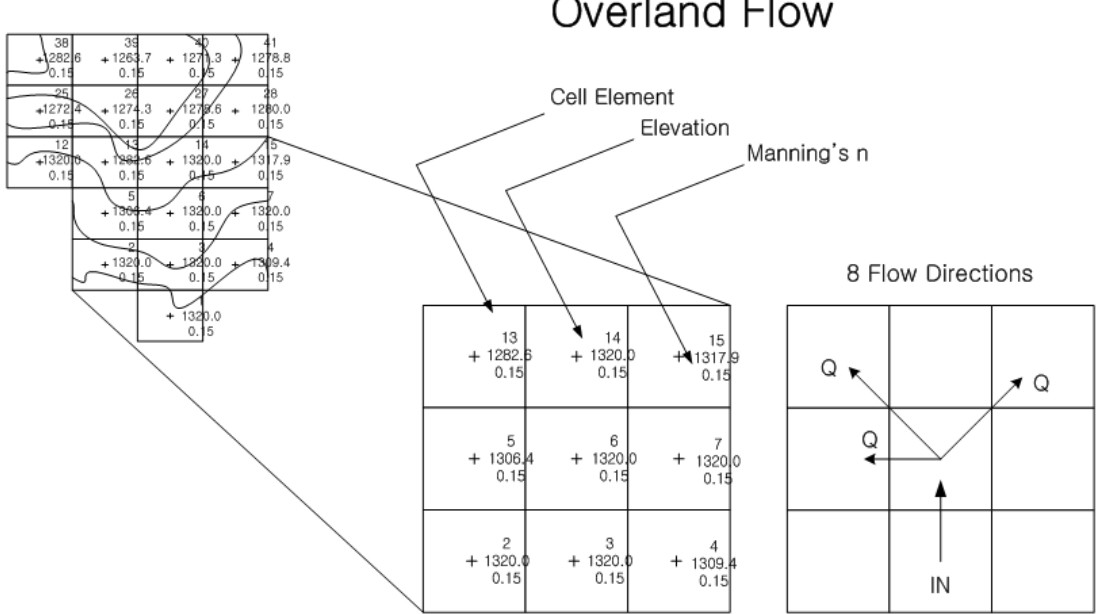

**Figure 4.** Discharge flux across cell element boundaries [34].

The Background and Review of Model Selection

The runoff analysis was conducted using the S-RAT model, which is a distributed rainfall–runoff model capable of cell unit analysis and has the advantage of precisely deriving the running time and leakage value because of its comprehensible interface. This model developed in Korea is now used in the research projects of the Korea Meteorological Administration [35].

On the other hand, the FLO-2D model is widely used in various studies in Korea [36–39]. However, in the case of models using shallow water equations (SWEs), such as FLO-2D, proper flood simulation is not possible because drainage is not considered [40]. In addition, there is a research trend where the flooding accuracy is low when the flooding simulation is performed using low-resolution data and is not suitable for a large-scale flooding simulation [41,42].

Therefore, the application of SWE [41,43–46] is appropriate if the analysis uses high-resolution data, as mentioned in Costabile et al. [47]. Furthermore, in resolving the problems of the previous FLO-2D, the discharge, considering the drainage capacity of the downtown area (5–15 years), was used as input data, and the topographic data of $30 \times 30$ m resolution was constructed and analyzed. In addition, it was judged that small-scale flooding analysis is more suitable for the flooding in Korea's urban areas because there are more local floodings compared to large-scale floodings [48].

*2.4. Study Area and Input Data Construction (Rainfall Data, S-RAT, and FLO-2D)*

2.4.1. Study Area

　　Seoul, Korea's capital, was selected as the target area for this study. Among Seoul's districts, the areas of Sadang-dong, Bangbae-dong, and Seocho-dong, which were affected by torrential rainfalls of approximately 300 mm from 1:00 to 23:00 on 27 July 2011, were divided into cells of 1 × 1 km for analysis. Figure 5 shows a picture of the target area divided into cells, and the analysis was conducted only for cells in Sadang-dong. On the other hand, Figure 6 is composed of a cell and inundation photographs of the actual damage on 27 July 2011, referring to the news, SNS, and past studies [49] at that time. Although many floods occurred, there was a limit in finding and verifying accurate literature regarding these events.

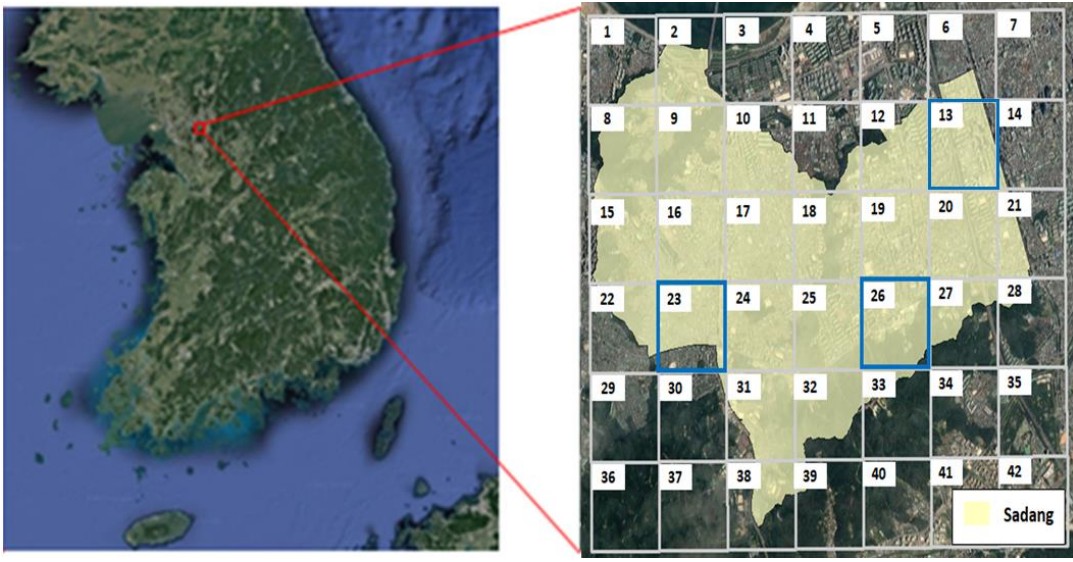

**Figure 5.** Study area.

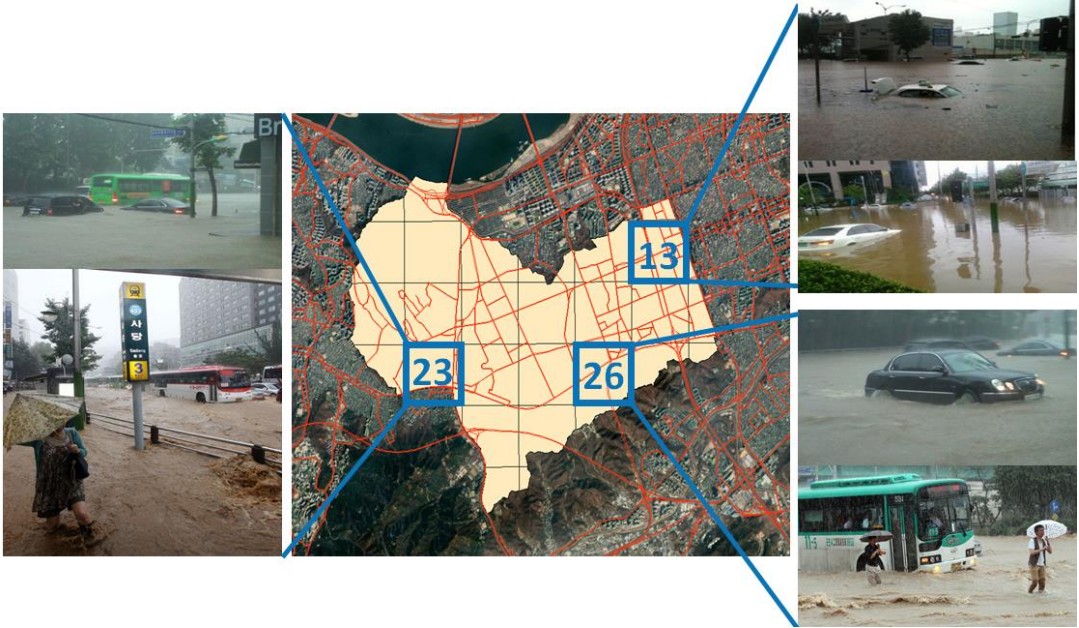

**Figure 6.** Photographs of actual damage to Sadang-dong area.

Figure 7 shows the cell numbers and road network of the Sadang-dong area and the speed limits for each cell. Most of the roads in Seoul are included, and the speed limit was 60 km/h. Meanwhile, Figure 8 is a hyetograph that shows rainfall events at the time of the event.

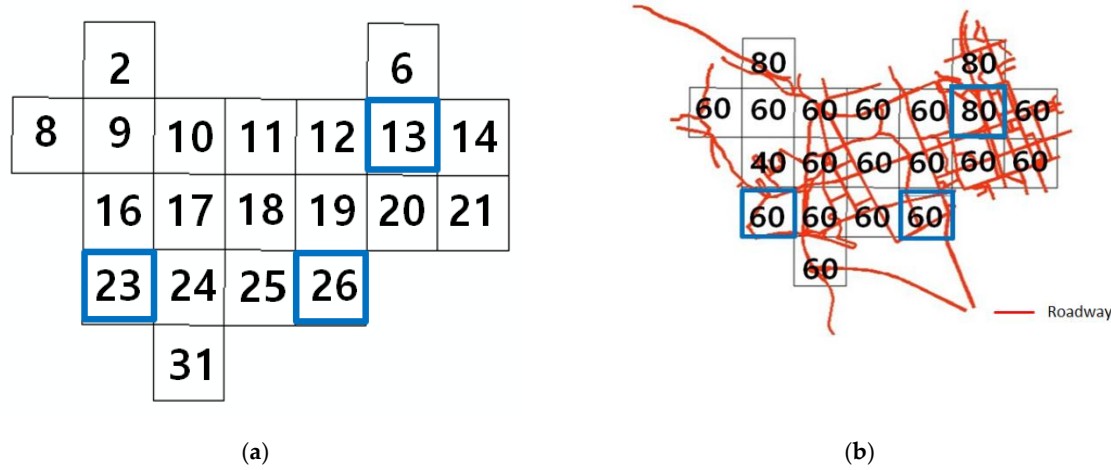

(**a**)                                                           (**b**)

**Figure 7.** Sadang-dong (**a**) cell number and (**b**) speed limit for each cell.

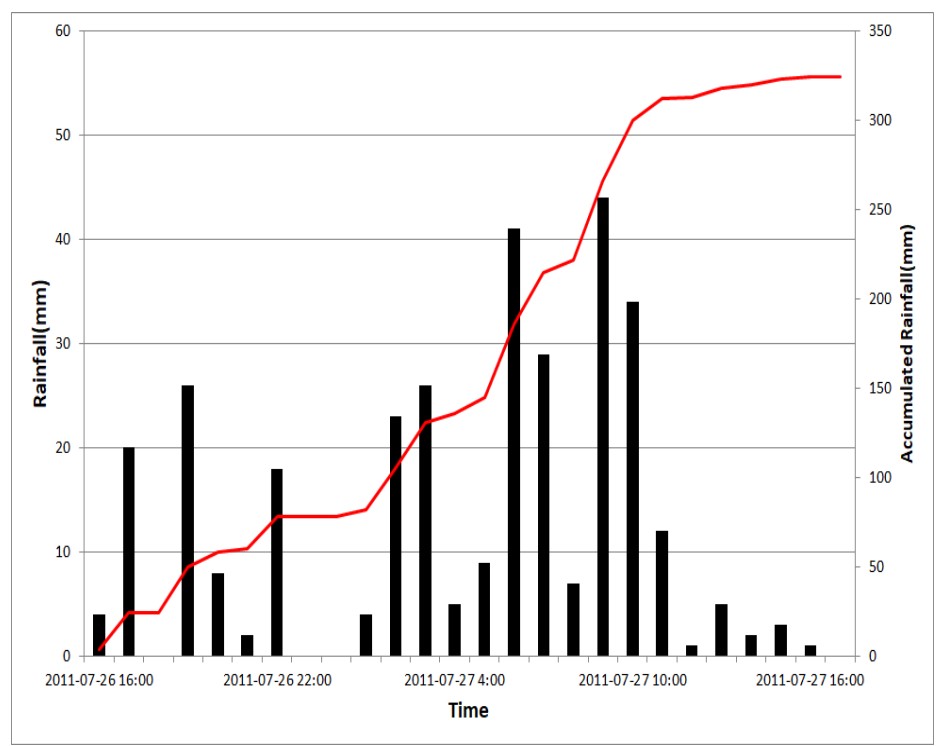

**Figure 8.** Hyetograph for Sadang-dong (26 July 2011, 16:00–27 July 2011, 16:00).

2.4.2. Input Data Construction

The rainfall–runoff model using S-RAT requires rainfall data and geomorphic data (Digital Elevation Model (DEM), land use, soil map). On the other hand, the Huff rainfall distribution method [50] was used with the third quartile, which is most used among Huff's quartiles of peak rainfall, to distribute the total rainfall in a time series. For rainfall data, 15 rainfall scenarios were set, with the 20–300 mm range divided into 20 mm units. The basic unit was 1 × 1 km, and the reference

point (inundation starting point) for applying the inundation model within the cell was defined as the lowest elevation point in the reference cell (1 × 1 km).

The rainfall data were input into the S-RAT model at intervals of 20 mm. For geomorphic data, drainage systems were formed using the SWAT tool in ArcGIS developed in USA, outlet points were created; subbasins were defined accordingly; and DEM, land use, and soil map were input.

Data necessary for the FLO-2D flooding analysis are discharge and DEM. Furthermore, discharge from S-RAT was applied to FLO-2D while spatial data in the same form of DEM were used in the rainfall–runoff model.

Figure 9 is the result of the Huff third-percentile rainfall distribution for 3 h, and Figure 10 is the construction data for DEM, land use, and soil map, which are input into the S-RAT model. The resolution of GIS topographic data used in this analysis was 30 × 30 m. For Figure 11, the result of dividing the watershed by the minimum elevation of each cell at Sadang Station exit is shown, with cells 13, 23, and 26 in red, indicating flood damage. The discharge computed here is the inland inflow rate of the flood inundation model and is used as the input value of flood analysis.

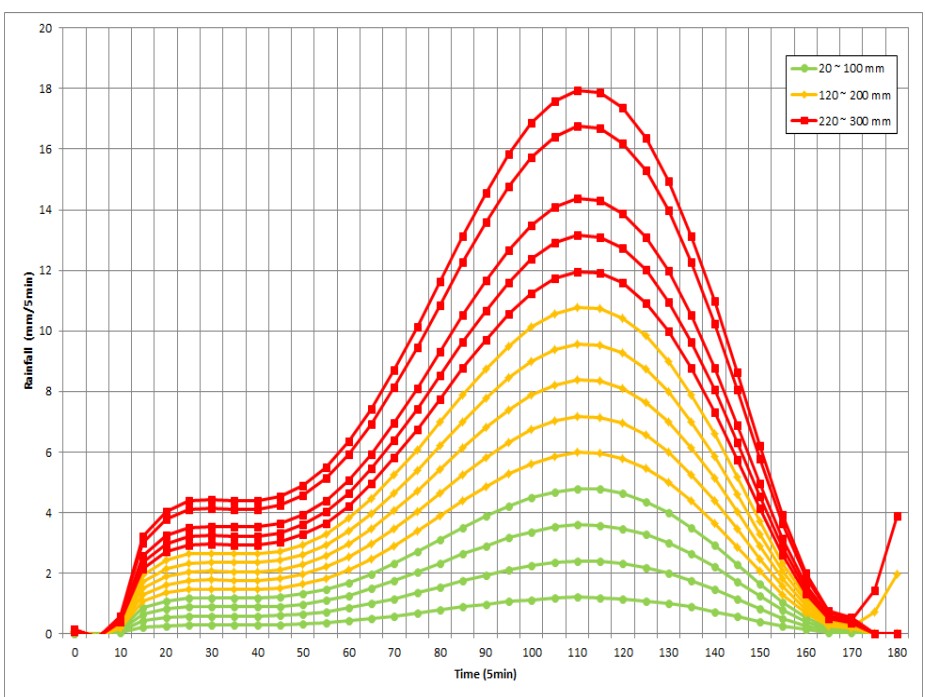

**Figure 9.** Rainfall scenarios with third quantile in 180 min.

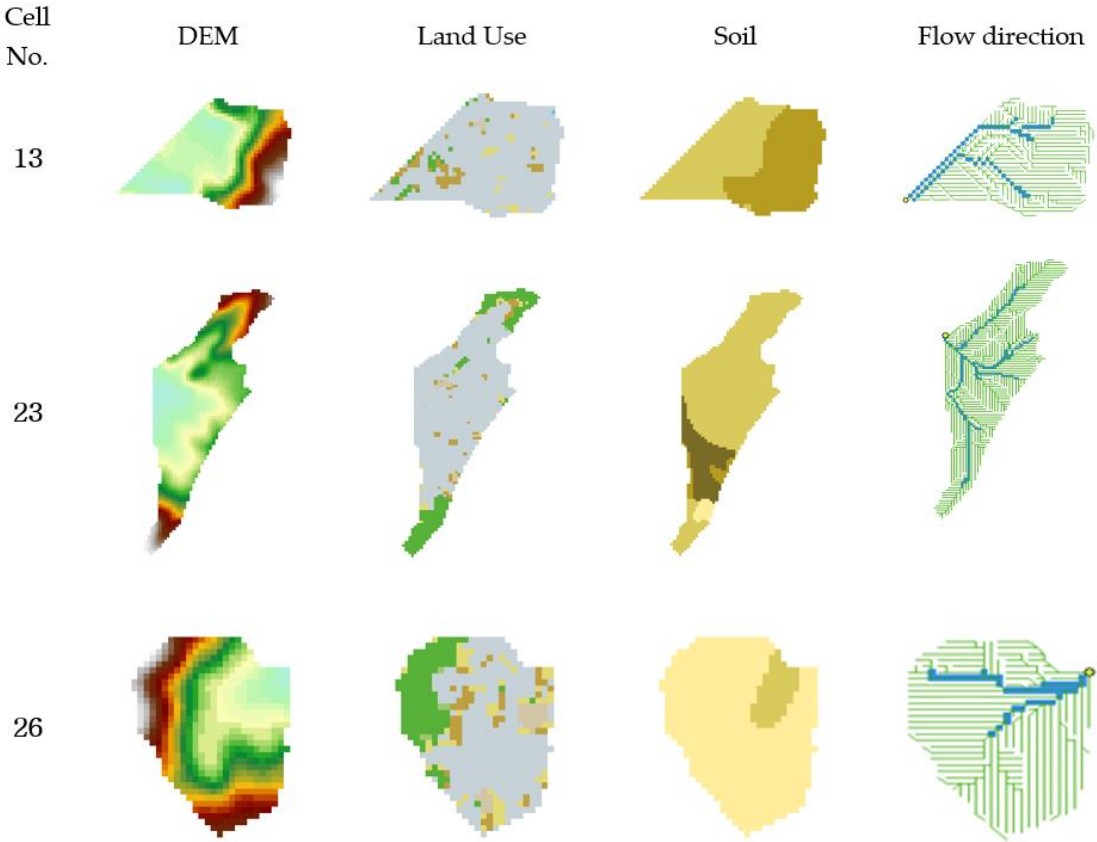

**Figure 10.** GIS (Geographic Information System) topographic data.

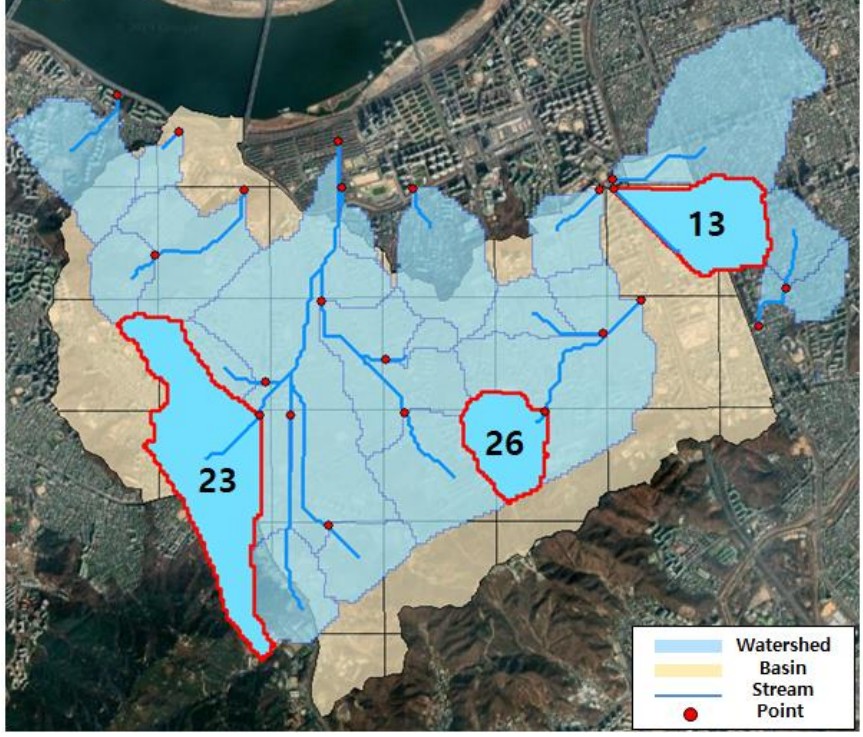

**Figure 11.** Basin for runoff model.

### 2.5. Rainfall–Depth Curve

The conceptual diagram of the rainfall–depth curve is shown in Figure 12. Rainfall of 20–300 mm was used with Huff's 3rd-quartile technique in the rainfall–runoff model, and discharge data according to rainfall were calculated by increasing rainfall at intervals of 20 mm. From the estimated flow rate, the flood depth data considering the flood protection capacity (5–15 years) is used as the input data for the flooding model. For a detailed analysis of the rainfall–depth curve that causes certain regional inundation, see Reference [35].

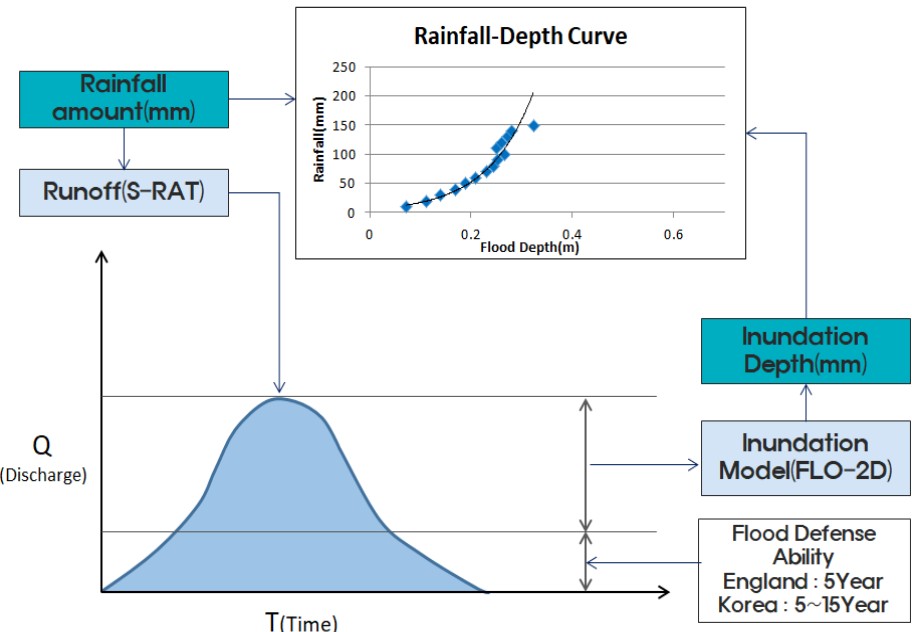

**Figure 12.** Conceptual diagram of rainfall–depth curve.

### 2.6. Depth–Vehicle Speed Curve

The depth–vehicle speed curve was selected using data for which prior analysis of flood depth and vehicle travel speed was completed using the relation of rainfall–depth–vehicle speed in this study. Table 1 shows the experimental data of [10], and the results are derived from static experiments and video data rather than statistically evaluating the vehicle speed when the road is actually flooded [51–58].

**Table 1.** Speed reduction caused by rainfall from previous studies.

| Point No. | Flood Depth (m) | Vehicle Speed (km/h) |
|:---:|:---:|:---:|
| 1 [51] | 0 | 88 |
| 2 [52] | 0.01 | 77 |
| 3 [53] | 0.087 | 40 |
| 4 [54] | 0.116 | 37 |
| 5 [55] | 0.125 | 26 |
| 6 [54] | 0.189 | 24 |
| 7 [56] | 0.200 | 16 |
| 8 [57] | 0.230 | 7 |
| 9 [58] | 0.250 | 3 |

In addition, in the previous study, vehicles of different sizes (small, large, and four-wheel drive) were classified, but the curves presented in this study refer to the average of each vehicle type. Moreover, most drivers are considered to drive above the speed limit at low flood depths. The regression equation was calculated by adjusting the speed up from the lowest value (Figure 13; Equation (8)).

$$y = 85 \times e^{-9x} (R^2 = 0.87) \tag{8}$$

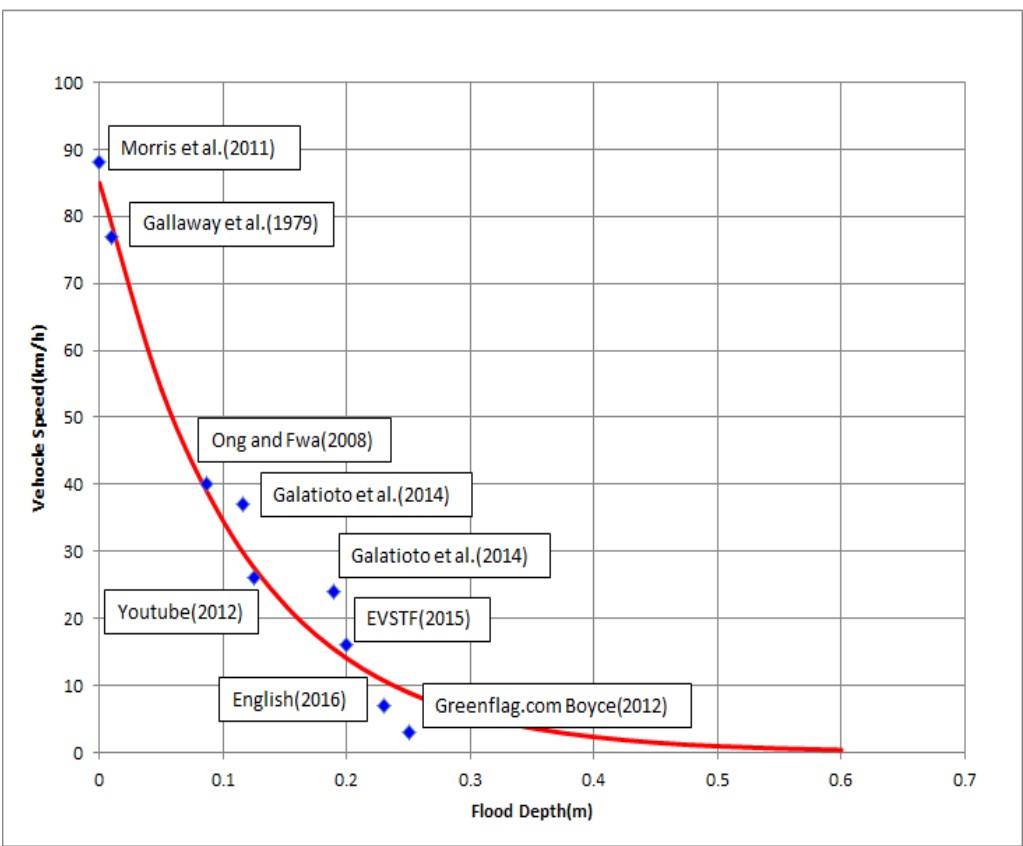

**Figure 13.** Estimation of depth–vehicle speed curve.

## 3. Results and Discussion

### 3.1. Results of the Application

The input data were applied to the flood model, and the results were obtained. Discharge computed at the lowest altitude point for cells 13, 23, and 26, the targets of analysis, is shown in Figure 14. Meanwhile, Figure 15 shows the results of computing the flood depth of the affected area based on the discharge of S-RAT using FLO-2D. In Figure 15, cell 13 showed a high flood depth, and the rainfall–depth curve was created based on the calculated depth of flooding.

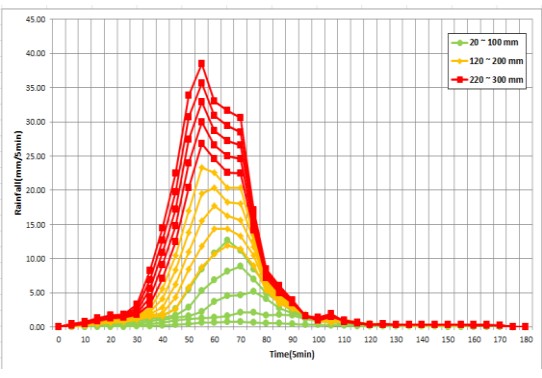

Cell 13

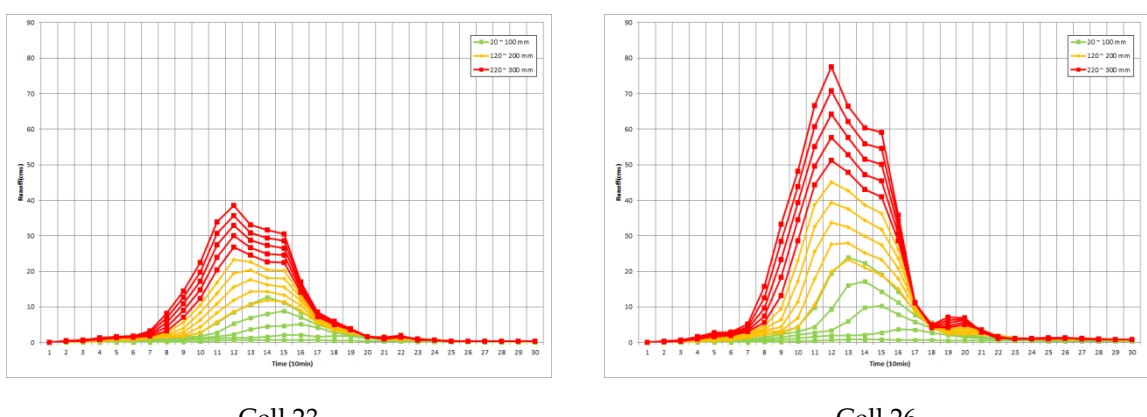

Cell 23　　　　　　　　　　　　　　　　　　　　　　　Cell 26

**Figure 14.** Result of runoff in the study area.

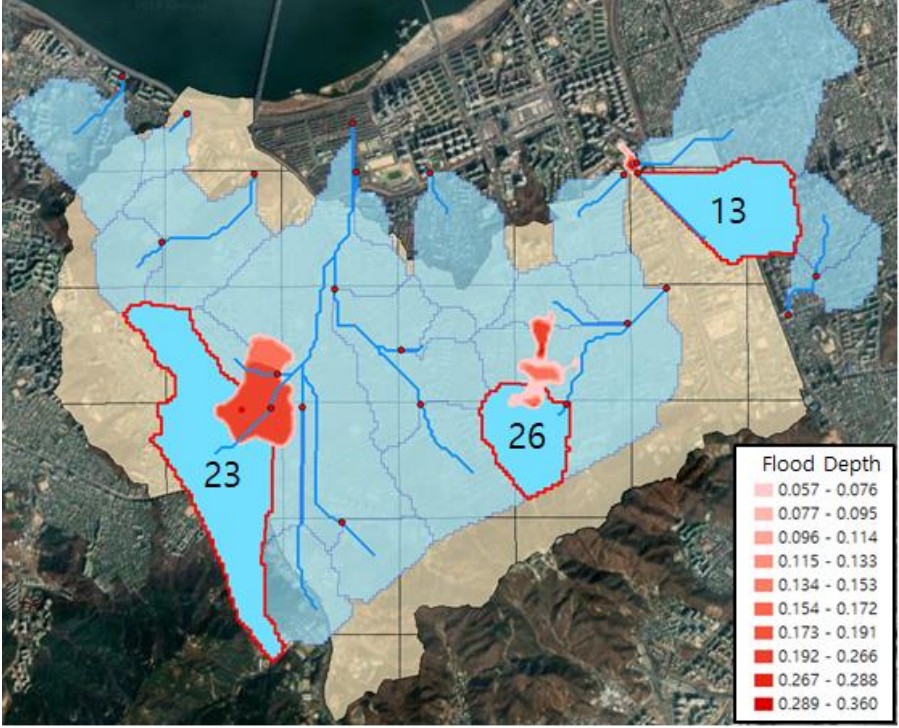

**Figure 15.** Flood depth result.

### 3.2. Creation of Rainfall–Depth Curve

In the rainfall–depth curve computation, the Sadang-dong area was divided into $1 \times 1$ km cells, and then, each was given a cell number and analyzed. This study intended to conduct an analysis in relation to vehicles, using a 20-cm exhaust port height for passenger cars and 1/2 of the tire height for trucks [30]. Table 2 shows the computed threshold rainfall of each cell at a flood depth of 20 cm applicable to the vehicle. Meanwhile, Figure 16 shows the rainfall–depth curve pertaining to cells 13, 23, and 26, where flooding actually occurred.

**Table 2.** Threshold rainfall of each cell.

| Cell No | Rainfall–Depth Curve | Threshold Rainfall Transport (20 cm) |
|:---:|:---:|:---:|
| 2,6 | $y = 26.91 \ln(x) + 78.641$ | 35.3 mm |
| 8 | $y = 485.34x - 17.749$ | 79.3 mm |
| 9,10 | $y = 296.96x - 11.816$ | 47.6 mm |
| 11 | $y = 24.67 \ln(x) + 88.98$ | 49.3 mm |
| 12 | $y = 2.6587e^{16.268x}$ | 68.8 mm |
| 13 | $y = 39.092x + 19.097$ | 26.9 mm |
| 14 | $y = 124.19x + 29.563$ | 54.4 mm |
| 16,17 | $y = 2.746e^{18.236x}$ | 105.4 mm |
| 18,19,20 | $y = 254.93x - 5.725$ | 45.3 mm |
| 21 | $y = 101.05x + 5.469$ | 25.7 mm |
| 23 | $y = 5.6508e^{11.133x}$ | 52.4 mm |
| 24 | $y = 11.677e^{11.589x}$ | 118.6 mm |
| 25 | $y = 470.82x - 36.811$ | 57.4 mm |
| 26 | $y = 554.51x - 32.897$ | 78.0 mm |
| 31 | $y = 4.6374e^{17.073x}$ | 132.8 mm |

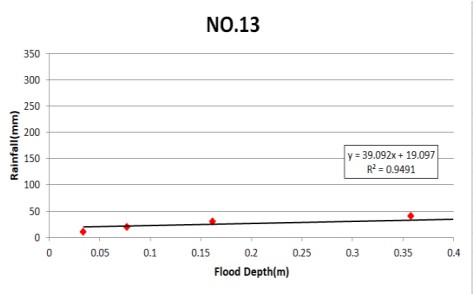

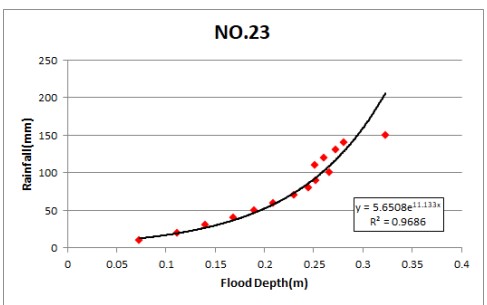
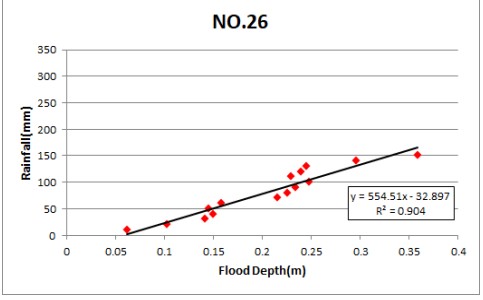

**Figure 16.** Rainfall–depth curve.

The height of the damaged image vehicle of cell 13, shown in Figure 6, is 1.475 m [59]. The measurement 1.353–1.623 m was derived when 70–80 mm, which is the maximum value for 3 h of cumulative rainfall, was added to the equation corresponding to cell 13. This depth is the height of the vehicle that is almost locked. When cells 23 and 26 are calculated, the flood depth of 0.23–0.245 m and 0.216–0.226 m can be derived, which is similar to the actual damage image showing a flood depth of up to half of the height of the bus wheel (0.49 m [60]).

*3.3. Results of Calculation by the Relationship between Rainfall–Flood Depth–Vehicle Speed*

Figure 17 shows the rainfall–depth–vehicle speed relations. On 27 July 2011, traffic control was implemented at 6:30 a.m. and lifted at 10:00 a.m. Analysis of the rainfall–depth–vehicle speed relation showed that vehicles could not pass the area at the time of damage.

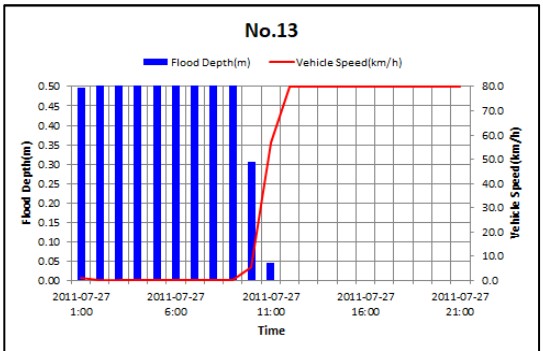

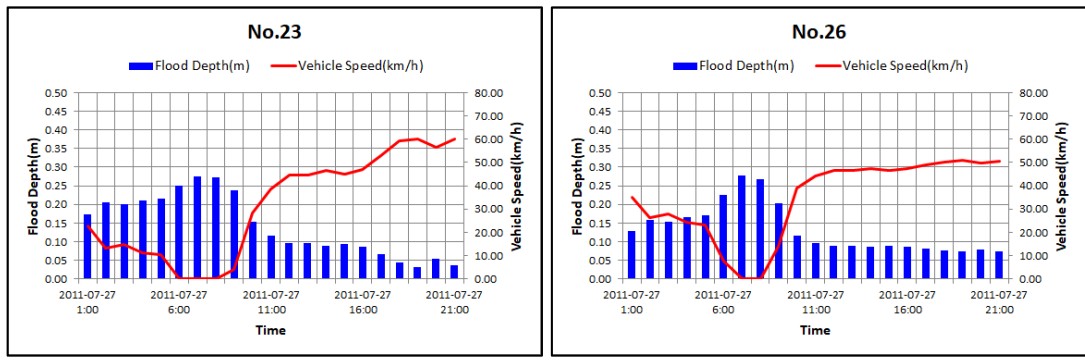

**Figure 17.** Result of rainfall–depth–vehicle speed relationship.

*3.4. Result of Calculation by Rainfall–Vehicle Speed Curve*

Using the estimated equation for vehicle travel speed reduction by rainfall from a previous article, vehicle travel speed was obtained by inputting actual rainfall events in the target area. Equation (9) estimates rainfall and vehicle travel speed reduction as suggested in a previous article [22]. The curve equation was computed using 5 min as a unit. Thus, to compare the previously suggested curve with the depth–vehicle speed curve of this paper, the cumulative rainfall for 3 h was divided into 5 min units and then it was compared with the 5 min rainfall by hour, showing the results in Figure 18. The hydrological characteristics of each cell failed to be reflected as just rainfall in the Seoul Special City was considered, and only the change caused by a speed limit in each cell can be analyzed.

$$y = 49.954x^3 - 88.924x^2 + 46.88x + 6.31 \ (R^2 = 0.70) \tag{9}$$

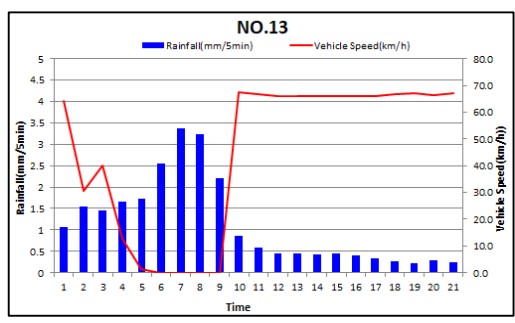

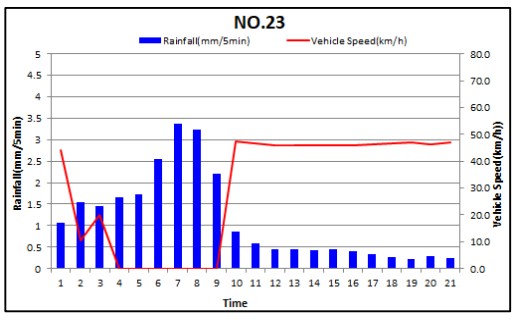
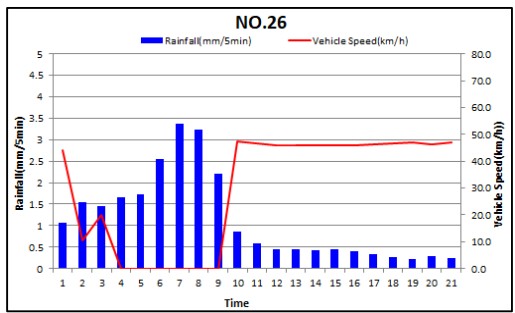

**Figure 18.** Rainfall–vehicle speed relationship.

*3.5. Traffic Disruption Simulation using Relationship of Rainfall–Depth–Vehicle Speed*

Using the rainfall–depth–vehicle speed relation computed, the reduction rate of vehicle travel speed for each cell was computed.

Figure 19 shows the change of vehicle speed according to hourly rainfall of cell 23. The time marked in red represents the hours of 06:00–13:00, when vehicle control was carried out on 27 July 2011. Based on this, the reduction rate grades were divided into four levels for the Sadang-dong area: very low, low, medium, and high. The division criteria into four levels was calculated following the reduction rate according to the speed limit for each cell, with very low at 0%–25%, low at 25%–50%, medium at 50%–75%, and high at 75%–100% (Figures 20 and 21). Figure 20a,b and Figure 21a,b are pictures of 07:00 and 09:00 at the time of control.

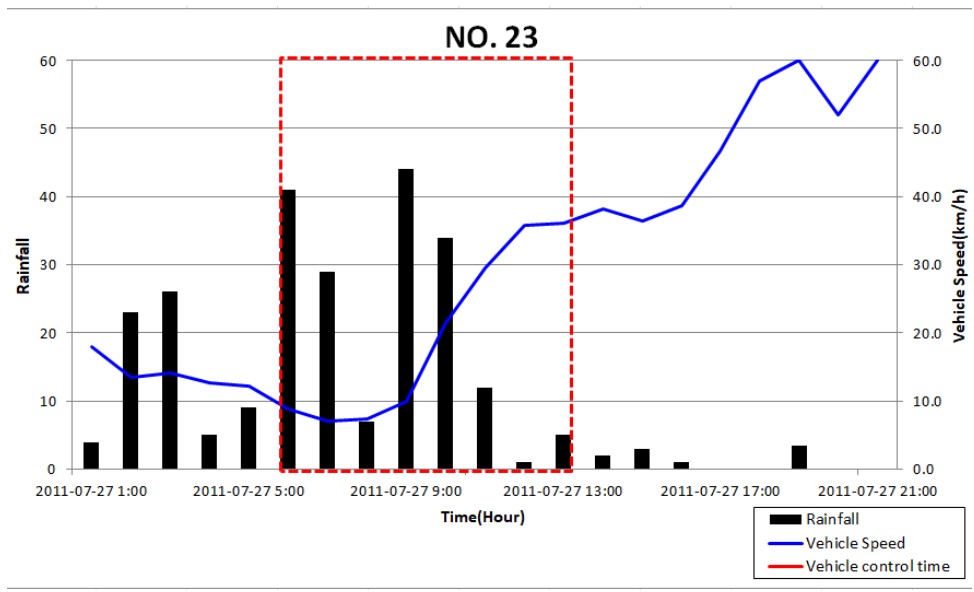

**Figure 19.** Vehicle speed variation over time.

In the existing literature, it is possible to check the time difference because there is no analysis by cell; however, information about the traffic by cell cannot be confirmed.

On the other hand, using the rainfall–depth–vehicle speed relations, the reduced rates of vehicle travel speed were identified owing to computation of flood depths for the cells during the hours of traffic control, and cells capable of smooth traffic despite traffic control were detected. It is judged that this method will help drivers reach their destinations by avoiding roads with high reduction rates of speed.

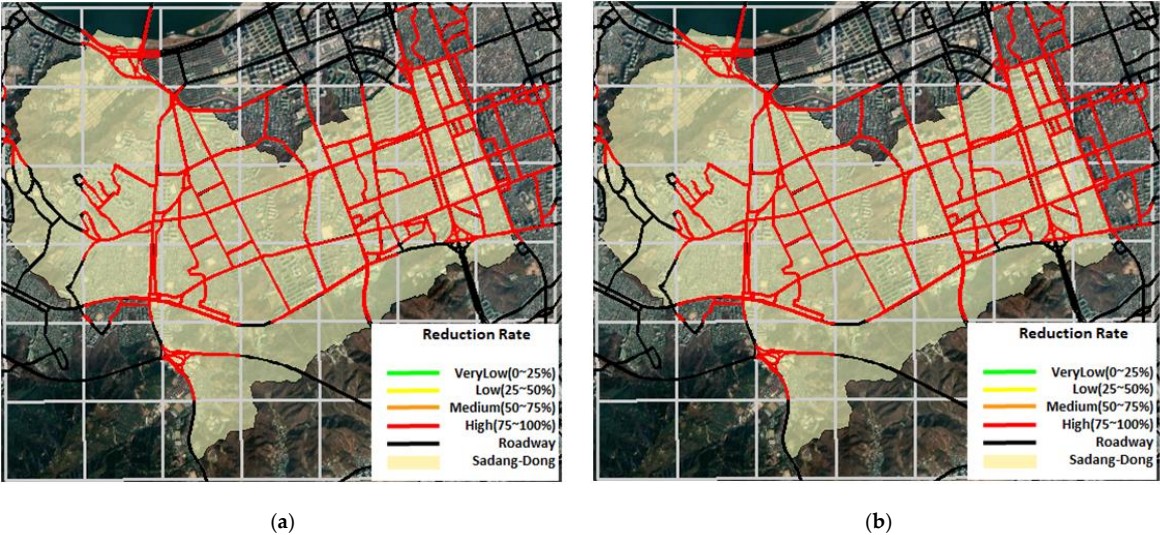

(**a**)          (**b**)

**Figure 20.** Traffic disruption map using rainfall–vehicle speed relationship. (**a**) Traffic Disruption Rating of 11.07.27 07:00, (**b**) Traffic Disruption Rating of 11.07.27 09:00.

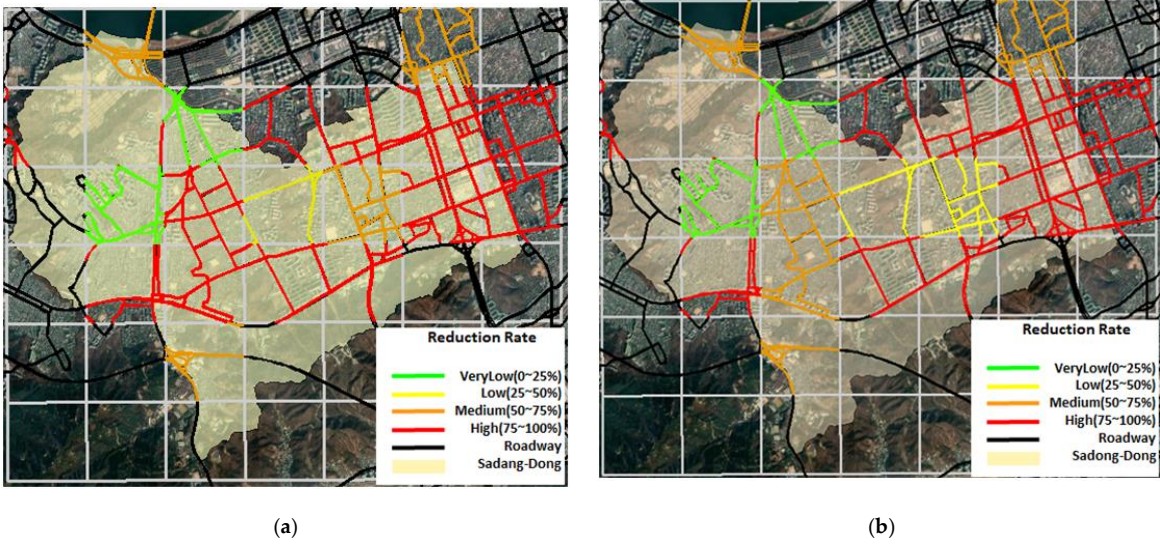

(**a**)          (**b**)

**Figure 21.** Traffic disruption map using rainfall–depth–vehicle speed relationship. (**a**) Traffic Disruption Rating of 11.07.27 07:00, (**b**) Traffic Disruption Rating of 11.07.27 09:00.

## 4. Conclusions

Damage from urban flooding should be one huge consideration in urban planning. Among many issues, urban areas continue to have a high vulnerability to short-term rainfall. In addition, anticipated climate change will increase the frequency of flooding, will negatively impact the economy, and will pose a threat to society's safety. Traditional road assessment methods were considered blocked



because of full flooding, so no proper study was conducted. Therefore, in this study, the risk of traffic disturbances caused by flooding in urban areas was to be assessed by classifying the reduction rate of vehicle speed in terms of impact forecasting.

In this paper, the target area was divided into a $1 \times 1$ km cell to obtain high-resolution results. The distribution runoff model S-RAT model was used for each cell, and the rainfall-flood depth curve was derived using the Flo-2D model, which is widely used in the shallow inundation simulation. The results were compared with previous study data and verified in the case of Sadang-dong, Seoul, on 27 July 2011, when actual damage occurred. The conclusion is as follows.

(1) Using this research method, the actual cases and the results of the analysis showed that the results of the depth of flooding using the rainfall–depth curve of the cell of 13,23,26 were mostly consistent with the depth of flooding of the actual damage photographs. In addition, based on the results of the analysis, it was found that the operation was almost impossible between 6:00 and 10:00 a.m. when the traffic restriction was implemented. Through this, we will be able to differentiate it from other studies by inundation analysis.

(2) The speed reduction rate was divided into four stages in the traffic disruption map. Therefore, as a result of comparative analysis with previous studies, only the difference in traffic effect according to time zone could be confirmed because the previous studies could not reflect hydrologic factors for each cell. However, the method presented in this paper was able to check the traffic information step by step by performing a hydrologic analysis for each cell.

For the purpose of the study, the speed limit of each road is assumed to be the maximum speed. In addition, weather phenomena other than rainfall were excluded and changes in driver behavior and restrictions on visibility caused by precipitation were not considered. In the case of discharge and flooding analysis, $30 \times 30$ resolution was analyzed but further analysis will be conducted using $5 \times 5$ data for precise analysis in the future.

Therefore, this study was found to be suitable for the early stage of the study of rainfall-flood depth-vehicle speed by cell, but many areas need to be improved. Although there was no flooding, there were cells exposed by flooding, which may be caused by various factors such as whether the drainage network or the driver's driving pattern is considered. Furthermore, it is necessary to improve the limit by comparing them with models that can be drainage considered in the future. In addition, the previous research data on the relationship between the flooding depth and the vehicle running speed is limited to the fact that the data are derived by static experiments not the statistics of the vehicle speed when the flood is actually inundated.

The rainfall–depth curve is currently calculated by simple linear regression, index, and logarithmic formula, but this has the disadvantage of releasing immersion values when rainfall values to increase. Afterward, we plan to use machine learning or nonlinear regression to advance the curved method. If the above limitations are supplemented, the study may serve as a useful framework for high-resolution forecasting, and rainfall forecasting may provide more accurate information than when reaching the destination in real time. In addition, the flood depth curve of this study can be used to select a point where the marginal rainfall is low and to apply it to flood prevention, road maintenance, and drainage projects.

**Author Contributions:** All the authors contributed to the conception and development of this manuscript. K.-S.C. and D.-H.K. carried out the survey of the previous study, wrote the graph of the data, and contributed to the search for actual damage data. B.-S.K. suggested idea of study and helped in analyzing the results and reviewed the writing. All authors have read and agreed to the published version of the manuscript.

**Funding:** This work was funded by the Korea Meteorological Administration Research and Development Program under Grant KMI (2018-03010).

**Conflicts of Interest:** The authors declare no conflict of interest.

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
