# Peer review of "Impact Assessment of Urban Flood on Traffic Disruption using Rainfall–Depth–Vehicle Speed Relationship"

_water, doi:10.3390/w12040926_

Round 1

Reviewer 1 Report

This is my second review of this work.

I regret to observe that several my comments have been completely neglected by the authors. I tried to help the authors in their revision in order to improve the quality of their work but I cannot find no significant modifications in the manuscript. So my major concerns on this work still hold.

The authors changed a lot their introduction but it still organized very badly. So I have to underline again what I've reported in my previous review: "The authors starts from the purposes of the work, presenting even the structure of the paper and then introduce the state of the art. This is the opposite of what is the common srtucture of the introductive section. Moreover, the state of the art is not really discussed and it seems to me a useless list of works without any critical review. In this way, it is quite hard to understand the gap in the literature and the advance of the proposed research."

Methodology section is still quite confused. No convincing replied have been provided in the new version of the manuscript to my previous comments. Moreover, I tried to help the authors focusing their attention on specific references that have been completely neglected by them. I report again my major comment:

"- I think that the authors should first explain the rationale
of their approach. Two models (FLO-2D and S-RAT) have been used to compute the rainfall threshold. No motivations have been used by the authors to motivate this choice. The results of the whole methodology
probably strongly depends on the choice of these two models.
- linked to the previous point, I have several concerns about the use of FLO-2D for a torrential flood event. As it is evident from equations (7), the FLO-2D solves the diffusive (or zero-inertia) approximations of the shallow
water equations (SWEs), neglecting inertial terms that could be very important in torrential events. In the literature, the use of 2-D SWEs is very common nowadays for this kind of event (see references [1], [2] and [3]
reported at the end of this review), and the use of the diffusive approximation can lead to significant errors (see [4]). I think that the use of FLO-2D is a major weak point of this paper and the authors should deeply
discuss its choice in the light of the references that I've recalled above."

Therefore, I cannot find any significant improvement in the new version of the manuscript. Considering the efforts made by the authors in their revision, my recommendation is "Major Revision". However, I strongly suggest the authors to deeply revise their work considering my comments and related literature.

cited works

[1] DOI: 10.1016/j.advwatres.2019.103392
[2] DOI: 10.1029/2018WR024083
[3] DOI: 10.1016/j.jhydrol.2016.03.021
[4] DOI: 10.1016/j.jhydrol.2017.11.033

Author Response

Response to Reviewer 1 Comments

» It is based on the file(water-730110(Without Tracking Function)) that is not applied to the tracking function.

Point 1: This is my second review of this work. I regret to observe that several my comments have been completely neglected by the authors. I tried to help the authors in their revision in order to improve the quality of their work but I cannot find no significant modifications in the manuscript. So my major concerns on this work still hold.

Response 1: I apologize for not making any revisions to the paper.

The authors changed a lot their introduction but it still organized very badly. So I have to underline again what I've reported in my previous review: "The authors starts from the purposes of the work, presenting even the structure of the paper and then introduce the state of the art. This is the opposite of what is the common structure of the introductive section. Moreover, the state of the art is not really discussed and it seems to me a useless list of works without any critical review. In this way, it is quite hard to understand the gap in the literature and the advance of the proposed research."

Response 1: The introduction, purpose, state of the art, and structure of the paper were modified. According to another reviewer’s suggestion, the structure of the paper was moved last, but it was deleted as it was unnecessary. Furthermore, we also added the review section (lines 128–144) about recent trends.

Point 2: Methodology section is still quite confused. No convincing replied have been provided in the new version of the manuscript to my previous comments.

Response 2: Another reviewer suggested that the headers in the table of contents be revised along with their content. The reviewer also recommended that I modify the composition to avoid confusion. In response to these recommendations, the methodology was added, followed by the description of the selected model, the reason for its selection, and the destination and input data, along with the concept of the curve (lines 154–313).

Point 3: Moreover, I tried to help the authors focusing their attention on specific references that have been completely neglected by them. I report again my major comment:

"- I think that the authors should first explain the rationale of their approach. Two models (FLO-2D and S-RAT) have been used to compute the rainfall threshold. No motivations have been used by the authors to motivate this choice. The results of the whole methodology probably strongly depends on the choice of these two models.

- linked to the previous point, I have several concerns about the use of FLO-2D for a torrential flood event. As it is evident from equations (7), the FLO-2D solves the diffusive (or zero-inertia) approximations of the shallow water equations (SWEs), neglecting inertial terms that could be very important in torrential events. In the literature, the use of 2-D SWEs is very common nowadays for this kind of event (see references [1], [2] and [3] reported at the end of this review), and the use of the diffusive approximation can lead to significant errors (see [4]). I think that the use of FLO-2D is a major weak point of this paper and the authors should deeply discuss its choice in the light of the references that I've recalled above."

Response 3: Section 2.2.1 was added to substantiate the model’s reason for selection and to provide the literature used as the basis for resolving the error (line 207–223). The resolution (30 m) used in this study is not mentioned in detail, so it is judged to have been caused confusion to be review. Afterward, I described the advantages and disadvantages of using shallow water equations (SWEs) in high resolution by referring to the papers that the reviewer suggested and presented the supporting points to address the limitations of the model.

Point 4: Therefore, I cannot find any significant improvement in the new version of the manuscript. Considering the efforts made by the authors in their revision, my recommendation is "Major Revision". However, I strongly suggest the authors to deeply revise their work considering my comments and related literature.

Response 4: Thank you for your detailed recommendations.

Reviewer 2 Report

An interesting paper aimed at assessing the impacts of urban floods on traffic disruption. I have some concerns that need to be addressed before accepting this manuscript.

1) The introduction section is disorganized. Everything related to the research gap and novely of the paper (lines 47-50) should be added after the literature review. If required (I doubt so), lines 51-53 should also be included at the end of the section. 

2) Please, include author's surname and year of publication in explicit citations(e.g. line 57).

3) The full meaning of acronyms (e.g. XAJ and SWAT in line 60) must be provided the first time they are used.

4) The cell size for the geographic inputs (DEM, Land use, etc.) is not provided, and this is extremely important in this kind of studies, especially in urban areas where resolutions must be very fine.

5) Section 2 should be named "Materials and methods", section 3 "Results and discussion" and section 4 "Conclusions".

6) I believe the calibration and validation of the methodology is not well approached. Flood simulation must be corroborated with real data. In this study, the authors limit this step to the selection of flood-prone cells according to past events, but they did not check whether the depth values calculated were consistent with those observed after real storms.

7) The conclusions are well addressed, including the main findings of the study and their implications, as well as pointing out the limitations and future lines of research to give continuity to it.

Author Response

Response to Reviewer 2 Comments

» It is based on the file(water-730110(Without Tracking Function) that is not applied to the tracking function.

Point 1: The introduction section is disorganized. Everything related to the research gap and novely of the paper (lines 47-50) should be added after the literature review. If required (I doubt so), lines 51-53 should also be included at the end of the section. 

Response 1: Lines 47–53 were moved to lines 139–144 at the end of the introduction, and the structure of the paper was deleted as it was unnecessary.

Point 2: Please, include author's surname and year of publication in explicit citations(e.g. line 57).

Response 2: I modified it by indicating the last names (lines 55–112).

Point 3: The full meaning of acronyms (e.g. XAJ and SWAT in line 60) must be provided the first time they are used.

Response 3: The full name of each model was provided as instructed (lines 60–61 and 213).

Point 4: The cell size for the geographic inputs (DEM, Land use, etc.) is not provided, and this is extremely important in this kind of studies, especially in urban areas where resolutions must be very fine.

Response 4: The resolutions of the geographic data to be studied were 30 × 30 m. and we will use 5 × 5 m for further research later. the relevant content was provided in lines 220–222 and 556–557.

Point 5: Section 2 should be named "Materials and methods", section 3 "Results and discussion" and section 4 "Conclusions".

Response 5: As the reviewer mentioned, Section 2 (line 150), Section 3 (line 399), and Section 4 (line 525) were modified, and the contents were added for each section. For Section 2, the methodology and materials were modified.

Point 6: I believe the calibration and validation of the methodology is not well approached. Flood simulation must be corroborated with real data. In this study, the authors limit this step to the selection of flood-prone cells according to past events, but they did not check whether the depth values calculated were consistent with those observed after real storms.

Response 6: It is difficult to obtain data on flood traces because it is past data in a downtown area. However, we have verified the depth of flooding through the news or photos from SNS at the time of the event. Moreover, the results of the reviewer’s analysis are included in lines 478–483 and 538–547.

Point 7: The conclusions are well addressed, including the main findings of the study and their implications, as well as pointing out the limitations and future lines of research to give continuity to it.

Response 7: Thank you for your careful review.

Round 2

Reviewer 1 Report

This is my third review of the work.

The authors have generally provided suitable answer to all my concerns.

I think that the paper has been significantly improved in respect to the previous versions. The overall presentation has reached now a sufficient level of quality. Methods are now discussed adequately and the results seems to be convincing.

Though I am not an English-native speaker, I think that the work requires a deep review of the English style.

For this reason, I suggest minor revision.

Author Response

Response to Reviewer 1 Comments

Point 1: This is my third review of the work. The authors have generally provided suitable answer to all my concerns. I think that the paper has been significantly improved in respect to the previous versions. The overall presentation has reached now a sufficient level of quality. Methods are now discussed adequately and the results seems to be convincing. Though I am not an English-native speaker, I think that the work requires a deep review of the English style. For this reason, I suggest minor revision.

Response 1: I think the quality has improved thanks to detailed reviews. I got an English editing service while modifying my review. So i'm going to attach the edit confirmation to the editor.
In addition, I've modified the grammar more.

Thank you!

Reviewer 2 Report

The authors are commended for their efforts to improve their manuscript according to my comments. I am satisfied with the new version of the document and think it is ready for publication.

Author Response

Response to Reviewer 2 Comments

Point 1: The authors are commended for their efforts to improve their manuscript according to my comments. I am satisfied with the new version of the document and think it is ready for publication.

Response 1: I think the quality has improved thanks to detailed reviews. I got an English editing service while modifying my review. So i'm going to attach the edit confirmation to the editor.
In addition, I've modified the grammar more.

Thank you!

This manuscript is a resubmission of an earlier submission. The following is a list of the peer review reports and author responses from that submission.

Round 1

Reviewer 1 Report

While the subject of the impact of rainfall in traffic flows should be of interest, the current paper is, in my opinion, not ready for publication for the following main reason:

This paper mainly operationalizes previously established relationships between rainfall & flood depth, and between flood depth & vehicle speed. No new insight is really gained. Although it is stated that “evaluation was conducted based on a real event”, it is not very clear from the study how this was done and whether it was conducted in a structured manner. There are no field observations from the 2011 rain event used for the analysis, and the presented evaluation seems to be based on very limited information about periods of traffic control.

Moreover, the paper seems somewhat un-developed in other areas:

The paper includes a rather elaborate literature study but it seems a bit un-reflected, superficial and with numerous unexplained acronyms.

Some passages are unclear: F.ex. what does it mean that “all grids show similar aspects with the existing rainfall-vehicle speed curve because the curve fails to reflect these characteristics” (29 + 270) ? And what are the implications?

The introduction 34-52 does not include any references. The underpinning of the presented general statements is unclear.

105-111 is actually about methodology and more or less repeated in the methodology section.

Figure 5 includes a number of labeled points without further explanation. F.ex. what does Youtube(2012) refer to?

Reviewer 2 Report

This paper deals with factors that influence vehicle travel speed, focusing in particular on flooding caused by rainfall. Specifically, the authors proposed a rainfall-depth curve to analyze change in travel speed during an inundation event caused by torrential rainfall.

The topic of the work is suitable for "Water". 

Though the paper has some points of novelty, I think that deep revisions and modifications are needed to reach a sufficient level of quality. In particular, the quality of presentation is very bad, several parts of the paper are unclear and some important assumptions are not motivated or discussed in the light of the more recent literature. 

In the following, I have highlighted several critical points of this work, in order to help the authors to improve the quality of their paper.

Abstract

- It is too long. There are at least a couple of ripetitions. Please remove that. 

- line 19: I do understand the meaning of [1]. If it refers to an equation, it should be removed because it is quite unusual to do that in the abstract. If it refers to a reference, it should be removed as well. 

Introduction: In my view, the introduction is organized very badly. The authors starts from the purposes of the work, presenting even the structure of the paper and then introduce the state of the art. This is the opposite of what is the common srtucture of the introductive section. Moreover, the state of the art is not really discussed and it seems to me a useless list of works without any critical review. In this way, it is quite hard to understant the gap in the literature and the advance of the proposed research.

Methodology:

 - line 114-115: "threshold rainfall inducing a particular flood depth in each region was calculated". This sentence is introduced without any motivation or explanation. What is this "particular" flood depth? What is its meaning? Same questions for the rainfall threshold. I think that the authors should first explain the rationale of their approach. two models (FLO-2D and S-RAT) have been used to compute the rainfall threshold. No motivations have been used by the authors to motivate this choice. The results of the whole methodology probably strongly depends on the choice of these two models.

- linked to the previous point, I have several concerns about the use of FLO-2D for a torrential flood event. As it is evident from equations (7), the FLO-2D solves the diffusive (or zero-inertia) approximations of the shallow water equations (SWEs), neglecting inertial terms that could be very important in torrential events. In the literature, the use of 2-D SWEs is very common nowadays for this kind of event (see references [1], [2] and [3] reported at the end of this review), and the use of the diffusive approximation can lead to significant errors (see [4]). I think that the use of FLO-2D is a major weak point of this paper and the authors should deeply discuss its choice in the light of the references that I've recalled above. 

- figure 3 is completely useless without explanations and comments. Moreover, the quality of the figure is very low.  

- section 2.4 : the description is too brief and unclear. Figure 4 is not discussed adequtely.

- section 2.5 : the whole section is quite unclear. Please add further comments and, in general, try to be more progressive.  

- section 2.6: same comments as before.

- figures 16, 17 and 18: how much do these results depends on the hydrodynamic models used? no discussion on model uncertainties is proposed by the authors.

- conclusions: I cannot find any general conclusions. The argumentations reported by the authors are too vague and in my view the strong limitations and uncertainties of this work are not underlined.

For these reasons, the paper cannot be accepted in the present form. Due to the amount of revisions, probably the best recommendation would be "reject but encouraging resubmission". However, I prefer to suggest major revision, with the specific request to provide detailed and carefully modifications of the work according to all my suggestions.

Minor Comments

 - Keywords: "unban flood" should be changed in "urban flood"

- lines 158-160: the variables should appear in Italics as in equation (7)

Cited works

[1] DOI: 10.1016/j.advwatres.2019.103392

[2] DOI: 10.1029/2018WR024083

[3] DOI: 10.1016/j.jhydrol.2016.03.021

[4] DOI: 10.1016/j.jhydrol.2017.11.033